# High-Dose Hydrocortisone Treatment Does Not Affect Serum C-Reactive Protein (CRP) Concentrations in Healthy Dogs

**DOI:** 10.3390/vetsci10100620

**Published:** 2023-10-15

**Authors:** Romy M. Heilmann, Niels Grützner, Peter H. Kook, Stefan Schellenberg, Jan S. Suchodolski, Joerg M. Steiner

**Affiliations:** 1Department for Small Animals, Veterinary Teaching Hospital, Faculty of Veterinary Medicine, Leipzig University, An den Tierkliniken 23, DE-04103 Leipzig, Germany; 2Ruminant and Swine Clinic, Free University of Berlin, Königsweg 65, DE-14163 Berlin, Germany; degruetz@web.de; 3Small Animal Clinic, Vetsuisse Faculty, University of Zurich, Winterthurerstrasse 260, CH-8057 Zurich, Switzerland; pkook@vetclinics.uzh.ch; 4Small Animal Specialty Center Aarau West, Muhenstrasse 56, CH-5036 Oberentfelden, Switzerland; s.schellenberg@tierklinikaw.ch; 5Gastrointestinal Laboratory, School of Veterinary Medicine and Biomedical Sciences, Texas A&M University, TAMU 4474, College Station, TX 77845, USA; jsuchodolski@cvm.tamu.edu (J.S.S.); jsteiner@cvm.tamu.edu (J.M.S.)

**Keywords:** autoinflammation, biomarker, canine, corticosteroid, immunosuppression, monitoring, steroid treatment

## Abstract

**Simple Summary:**

C-reactive protein (CRP) can be measured in serum and is a useful biomarker to diagnose inflammatory conditions (e.g., autoimmune diseases) and monitor the response to treatment in dogs. Because dogs with autoimmune diseases are often treated with high doses of glucocorticoids, a potential effect of this treatment on serum CRP concentrations—apart from the disease process—cannot be excluded. To address this knowledge gap, we measured serum CRP concentrations in five different time points in healthy dogs that were given either a high dose of a glucocorticoid (hydrocortisone) or a placebo over 28 days. We observed fluctuations in serum CRP concentrations in individual dogs during the study period, but the treatment did not significantly influence these fluctuations. Thus, confounding effects of corticosteroid administration on the interpretation of serum CRP concentrations as an inflammatory biomarker in dogs is unlikely, but this warrants confirmation by further studies.

**Abstract:**

Measuring C-reactive protein (CRP) in serum is a useful surrogate marker for assessing disease progression and treatment response in dogs with autoinflammatory diseases. Affected dogs often receive high-dose glucocorticoid treatment, but the effect of such treatment alone on serum CRP concentrations is unknown. We evaluated serum CRP concentrations via immunoassay (sandwich enzyme-linked immunosorbent assay and particle-enhanced turbidimetric immunoassay) in 12 healthy beagle dogs administered high-dose hydrocortisone (8 mg/kg q12 h) per os vs. placebo over 28 days (days 0, 1, 5, and 28) in a randomized parallel study design. Serum CRP concentrations slightly decreased during treatment or placebo but without a significant association with hydrocortisone administration (*p* = 0.761). Compared to baseline, serum CRP concentrations were decreased by >2.7-fold (minimum critical difference) in three hydrocortisone-treated dogs and two dogs in the placebo group on day 28, whereas an increase to >2.7-fold was seen in one dog receiving placebo. These results suggest a lack of confounding effects of high-dose hydrocortisone administration on serum CRP concentrations in healthy dogs. This might also hold in dogs with autoinflammatory conditions and/or administration of other high-dose corticosteroids, suggesting that CRP presents a suitable biomarker to monitor inflammatory disease processes. However, this needs confirmation by further studies evaluating corticosteroid-induced cellular (e.g., hepatic) transcriptome and proteome changes.

## 1. Introduction

C-reactive protein (CRP) is a positive acute-phase protein of the pentraxin family that increases in serum with systemic inflammation, infection, and cancer [1]. CRP production by the liver responds to interleukin (IL)-6 and IL-1β and is regulated by nuclear factor kappa-B (NF-κB) [2]. Measurement of serum CRP is clinically useful as a surrogate marker to assess disease progression and response to treatment in dogs with autoinflammatory conditions [3,4]. Serum CRP concentrations might also distinguish dogs with chronic inflammatory enteropathy (CIE) requiring anti-inflammatory or immunosuppressant medication from food- or antibiotic-responsive cases [5].

Dogs with autoinflammatory diseases such as idiopathic inflammatory bowel disease [4], primary immune-mediated hemolytic anemia [6], steroid-responsive meningitis-arteritis [7], or primary immune-mediated polyarthritis, where serum CRP is used for patient monitoring [3], often receive anti-inflammatory or immunosuppressive doses of glucocorticoids as a first-line treatment.

In people, it has been suggested that corticosteroids can decrease serum CRP concentrations without an improvement or any change in the underlying disease process that caused serum CRP concentrations to increase [8]. However, distinguishing clinical improvement of a disease process from an independent steroid-induced decrease in CRP expression is difficult—particularly in clinical patients—and the possibility of an effect of such treatment alone on serum CRP concentrations in dogs is currently unknown.

Data on the existence of such an effect, or lack thereof, are an important prerequisite to assess the utility of serial serum CRP measurements as a monitoring tool in clinical practice [1,9,10]. Thus, we aimed to evaluate the possibility of an effect of high-dose corticosteroid treatment over 4 weeks on serum CRP concentrations in healthy dogs.

Several assay formats can be used to measure serum CRP in specimens from dogs, which include species-specific enzyme-linked immunosorbent assays (ELISA), immunoturbidimetric assays, and a patient-side lateral flow immunoassay [8,11,12,13]. Thus, a secondary aim was to compare the results obtained using two species-specific canine serum CRP assays.

## 2. Materials and Methods

Samples from a randomly allocated, prospective experimental study population of 12 healthy 3-year-old beagle dogs with iatrogenic hypercortisolism were used. Clinical and other biomarker data were reported for these dogs elsewhere [14,15,16], showing that hypercortisolism was reliably induced in the hydrocortisone-treated group [14,16]. The protocol for collecting serum samples was approved by the regional authority (Cantonal Committee for the Authorization of Animal Experimentation, Zurich, ZH, Switzerland; approval no. 150/2004, approved on 6 August 2004).

Serum samples were collected from each dog before initiating treatment (baseline, day 0) and on days 1, 5, and 28 of receiving either hydrocortisone (Hotz Pharmacy, Kusnacht, Switzerland; 8 mg/kg PO q12 h; *n* = 6) or placebo (gelatin capsule; *n* = 6) [14]. The samples were stored at −80 °C for 96 months until serum CRP analysis using the TriDelta sandwich ELISA (Tri-Delta Phase CRP, Tri-Delta Diagnostics, Boonton Township, NJ, USA) with a reference interval (RI) of 0.1–7.6 mg/L [13], then stored for another 96 months until being tested on the Gentian canine CRP assay (Gentian Canine CRP particle-enhanced turbidimetric immunoassay, Gentian Diagnostics, Moss, Norway; RI: <10 mg/L). This was considered reasonable given the long-term stability (at least 11 years) of serum CRP reported for human sera stored at −80 °C [17]; the same aliquots of samples were used for both measurements. Serum CRP concentrations were measured in all samples over 2 assay runs for the TriDelta assay (96-well plate format) and 1 assay run for the Gentian assay (on a laboratory chemistry analyzer) to reduce inter-assay variability using the same reagents, calibrators, and quality control solutions.

Normality and equality of variances of the data were tested using a Shapiro–Wilk *W* test and a Brown–Forsythe test. Correlation and agreement between the results obtained with the TriDelta and Gentian assay were evaluated via calculating a Spearman correlation coefficient *ρ*, Bland–Altman plot (using common log-transformed data), and kappa statistic (using the upper reference limits of each assay as decision limits). BoxCox transformed (λ = −0.93) data obtained using the TriDelta assay were analyzed using a MANOVA model with a repeated measures design, followed by t-tests to detect differences between both groups of dogs at each time point. JMP (v13.0, SAS Institute, Cary, NC, USA) and GraphPad Prism (v9.0 and 10.0, GraphPad Software, San Diego, CA, USA) software were used for statistical analyses, and *p* < 0.05 indicated statistical significance.

## 3. Results

Serum CRP concentrations were above the lower detection limit (LOD) of the assay in 17 samples (35%) with the TriDelta assay (LOD = 0.1 mg/L) and 3 samples (6%) with the Gentian assay (LOD = 9.9 mg/L). Concentrations of CRP measured above the RI of the TriDelta assay (>7.6 mg/L) and the Gentian assay (≥10 mg/L) in 3/48 (6%) serum samples (one in each group at baseline and one dog on day 1 of treatment). Results obtained using these assays for the 48 serum samples were moderately correlated (Figure 1). The difference in common log-transformed serum CRP concentrations between both assays plotted against their mean (Bland–Altman plot) yielded slightly lower results on the TriDelta assay (Figure 2), with a mean difference (bias) of −3.49, 95%CI: −3.97–−3.01 (log scale; corresponding to 0.03 mg/L, 95%CI: 0.02–0.05 mg/L); the lower and upper limits of agreement were −6.71, 95%CI: −7.53–−5.89 (log scale; corresponding to 0.01 mg/L, 95%CI: 0–0.01 mg/L) and −0.27, 95%CI: −1.09–0.55 (log scale; corresponding to 0.76 mg/L, 95%CI: 0.34–1.74 mg/L), respectively.

Serum CRP concentrations slightly decreased over the 28 days of treatment or placebo (Figure 3), without a significant effect of hydrocortisone administration detected (*p* = 0.761; MANOVA). High biological variation was previously shown for serum CRP concentrations in healthy dogs, and the minimum critical difference (MCD) for clinically relevant changes (decrease or increase) in serial CRP measurements was reported as >2.7-fold [18]. Compared to baseline (day 0) measurements, serum CRP concentrations on day 1 of treatment decreased by >MCD in two hydrocortisone-treated dogs and one dog in the placebo group and increased by >MCD in two dogs in each group. In four hydrocortisone-treated and two placebo-receiving dogs, serum CRP concentrations decreased >MCD from day 1 to day 5. From day 5 to day 28, serum CRP concentration decreased >MCD in one hydrocortisone-treated dog and two dogs in the placebo group, whereas in one placebo-given dog, an increase >MCD was seen.

## 4. Discussion

The results of our study suggest a lack of a significant confounding effect of high-dose hydrocortisone administration on serum CRP concentrations in healthy dogs. This might also hold in dogs with primary inflammatory conditions where serum CRP concentration is used to monitor the disease and the response to treatment and/or in dogs receiving high doses of other (more commonly used) corticosteroids. Showing that serum CRP concentrations are not significantly affected by such high doses of a corticosteroid, presenting the most common first-line immunosuppressive treatment in dogs, validates the clinical utility of the CRP test in monitoring dogs with inflammatory conditions. A similar investigation in affected dogs would be needed to decipher the effects of corticosteroid treatment on serum CRP concentrations from the immunomodulatory effects of treatment on the disease reflected by a decrease in serum CRP concentration [19,20]. However, such a study would be very challenging, particularly for ethical reasons, because immunosuppressive doses of corticosteroids often cannot be withheld in affected dogs during the induction phase of therapy without inflating the risk of a poor response to treatment and/or unfavorable patient outcomes.

Despite being only moderately correlated, the results obtained using the TriDelta and the Gentian assay showed good agreement, but only the TriDelta assay allowed readings within the RI. A high-sensitivity canine CRP assay might have yielded serum CRP measurements in all dogs but is currently not widely or routinely available. A limitation is that serum samples stored at −80 °C for up to 192 months were used for CRP measurement in this study, but CRP is reported to be highly stable in serum samples [17]. Repeated CRP measurement using the TriDelta assay, which has a lower detection limit of 0.1 mg/L (compared to 10 mg/L of the Gentian assay), at the time of re-testing using the Gentian assay would have been preferred but was not possible due to lack of assay availability. Serum CRP concentrations also agreed with previous data in healthy beagle dogs utilizing a different immunoassay [21]. Furthermore, hydrocortisone was used to induce iatrogenic hypercortisolism in the dogs included in the study rather than prednisone or prednisolone, which are more commonly used to treat dogs. However, similar (if any) effects of these drugs on CRP concentrations can be reasonably expected, given that the equivalent to an immunosuppressive dose of prednisolone was used in this study and from the evidence of cortisol excess in these dogs.

## 5. Conclusions

The results of this study show that high-dose hydrocortisone administration does not confound serum CRP concentrations in healthy dogs. This might also hold in dogs with autoinflammatory conditions and administration of other high-dose corticosteroids, which suggests that serum CRP is useful for monitoring inflammatory disease processes. Further research is warranted to study corticosteroid-induced cellular transcriptome and proteome changes in dogs with or without primary inflammatory conditions.

## Figures and Tables

**Figure 1 vetsci-10-00620-f001:**
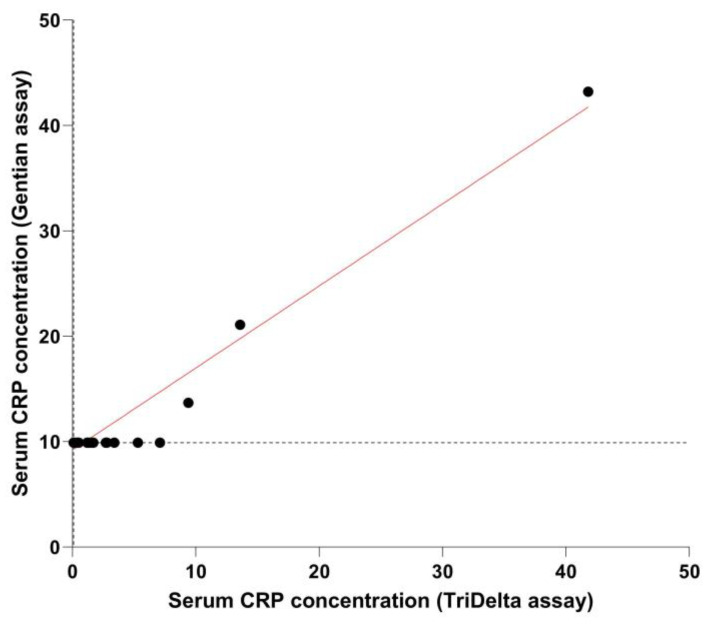
Serum canine CRP concentrations measured by two different assay systems. The results obtained using the TriDelta canine CRP assay (x-axis; assay lower detection limit: 0.1 mg/L, vertical dashed line) and the Gentian canine CRP assay (y-axis; lower detection limit: 9.9 mg/L, horizontal dashed line) for the 48 serum samples from 12 dogs (4 time points during treatment with placebo or hydrocortisone at 8 mg/kg q12 h for 28 days; black dots) were only moderately correlated (*p* = 0.49, 95%CI: 0.23–0.69; *p* < 0.001; red line indicates linear correlation). However, evaluation of the diagnostic agreement between paired results from both assays yielded a kappa coefficient of 1.00 (95%CI: 1.00).

**Figure 2 vetsci-10-00620-f002:**
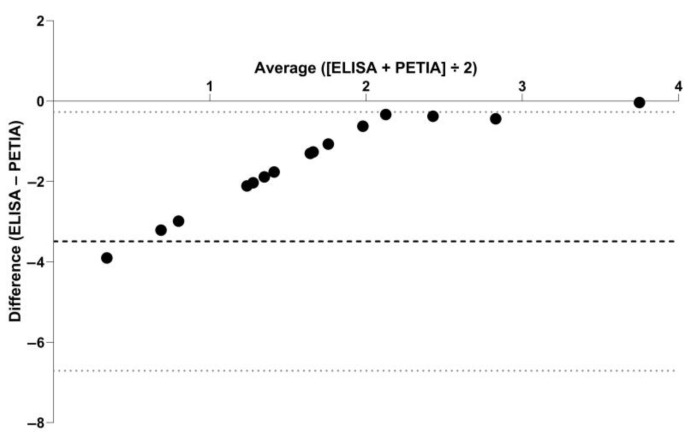
Bland–Altman plot showing the agreement of serum CRP concentrations in 48 serum samples measured by sandwich enzyme-linked immunosorbent assay (ELISA) and particle-enhanced turbidimetric immunoassay (PETIA). Symbols (black dots) represent the difference between both methods (difference of ELISA–PETIA) against their mean (average concentration calculated for ELISA and PETIA) for the individual serum samples. The mean difference (bias; black dashed horizontal line) and the lower and upper limit of agreement (gray dotted horizontal lines) were calculated as −3.49, −6.71, and −0.27 (common log scale; corresponding to 0.03 mg/L, 0.01 mg/L, and 0.76 mg/L).

**Figure 3 vetsci-10-00620-f003:**
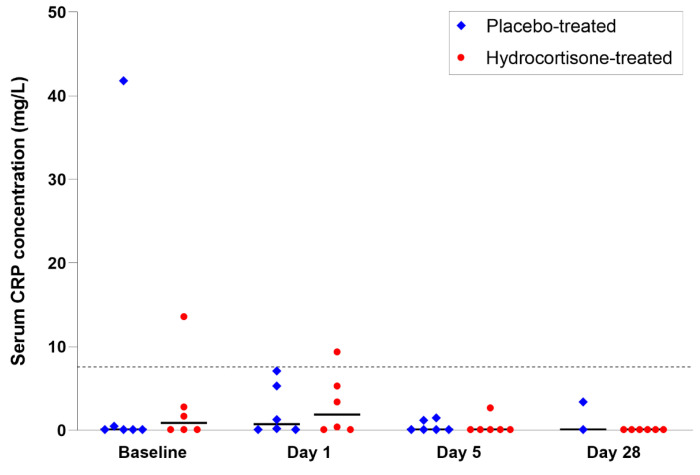
Serum CRP concentrations in healthy dogs given hydrocortisone (*n* = 6) vs. placebo (*n* = 6). Two baseline samples (one in each group of dogs) and one sample obtained on day 1 of hydrocortisone treatment yielded serum CRP concentrations above the upper reference limit of the TriDelta CRP assay (>7.6 mg/L; gray dashed line); all other measurements were within the RI. Serum CRP concentrations slightly decreased over the 28 days of treatment in both groups of dogs, but there was no significant effect of hydrocortisone administration on the concentrations detected (*p* = 0.761). Black horizontal lines indicate the medians for each group and time point.

## Data Availability

Data are available from the first or second author upon reasonable request.

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
