# Peer review of "High-Dose Hydrocortisone Treatment Does Not Affect Serum C-Reactive Protein (CRP) Concentrations in Healthy Dogs"

_vetsci, 2023, doi:10.3390/vetsci10100620_

Round 1

Reviewer 1 Report

Include the bland Altman plots and explain them. 

Overall research an manuscript is good.

Author Response

  1. Reviewer comment: Include the bland Altman plots and explain them.

Author response: We thank the reviewer for this remark and have added the Bland-Altman plot as an additional figure (Fig. 2 and lines 135-142).

  1. Reviewer comment: Overall research and manuscript is good.

Author response: We thank the reviewer for the careful review and the overall very positive assessment of our manuscript.

Reviewer 2 Report

As a peer reviewer for this journal, I have carefully examined the paper and have some concerns that warrant consideration before a final decision is made regarding its acceptance.

Firstly, I would like to address the methodology employed in the study. While the treatments and experimental design appear to be adequately described, there is an aspect that requires clarification. Specifically, it remains unclear whether the MANOVA (Multivariate Analysis of Variance) applied in the analysis accounts for paired groups, given that the study examines CRP values within the same dogs at different time points. If the analysis utilized a test designed for independent groups rather than paired groups, this could potentially impact the validity and interpretation of the study's outcomes. Therefore, I recommend seeking further clarification and confirmation regarding the appropriateness of the statistical approach chosen for the paired data.

Secondly, the authors state that they induced iatrogenic hypercortisolism; however, they do not present any serological or clinical analyses to substantiate this claim. The absence of such supporting data raises questions about the verifiability of the induced condition. To strengthen the paper's scientific rigor, I suggest that the authors consider providing relevant serological or clinical evidence to confirm the induction of iatrogenic hypercortisolism in the experimental subjects.

Furthermore, I believe it is essential to address the clinical implications of the study's findings. Specifically, if the authors assert that CRP levels do not significantly vary with hydrocortisone treatment, it would be valuable to discuss the potential clinical impact of this observation. Does this result have any implications for the management of chronic inflammatory conditions, or does it suggest alternative therapeutic approaches? If so, it would greatly enhance the practical relevance of the research if the authors could elaborate on these aspects in the discussion section.

In conclusion, while the manuscript demonstrates promise and contributes to the scientific discourse, these concerns regarding the statistical analysis, the substantiation of induced hypercortisolism, and the clinical implications need to be addressed comprehensively. I believe that with the authors' cooperation in addressing these issues, the manuscript could be significantly strengthened and warrant consideration for publication.

a

Round 2

Reviewer 2 Report

a

a